# Inflammatory responses in SARS-CoV-2 associated Multisystem Inflammatory Syndrome and Kawasaki Disease in children: An observational study

G. Biesbroek[1]*, B. Kapitein[2°], I. M. Kuipers[3°], M. P. Gruppen[1], D. van Stijn[1], T. E. Peros[2], M. van Veenendaal[4], M. H. A. Jansen[4], C. W. van der Zee[5], M. van der Kuip[1], E. G. J. von Asmuth[6], M. G. Mooij[6], M. E. J. den Boer[7], G. W. Landman[8], M. A. van Houten[9], D. Schonenberg-Meinema[1], A. M. Tutu van Furth[1], M. Boele van Hensbroek[1], H. Scherpbier[1], K. E. van Meijgaarden[10], T. H. M. Ottenhoff[10], S. A. Joosten[10], N. Ketharanathan[11], M. Blink[12], C. L. H. Brackel[5,13], H. L. Zaaijer[14], P. Hombrink[15], J. M. van den Berg[1], E. P. Buddingh[6‡], T. W. Kuijpers[1,15‡]

1 Department of Pediatric Immunology, Rheumatology and Infectious Disease, Emma Children's Hospital, Amsterdam UMC Location University of Amsterdam, Amsterdam, The Netherlands, 2 Pediatric Intensive Care Unit, Emma Children's Hospital, Amsterdam UMC Location University of Amsterdam, Amsterdam, The Netherlands, 3 Pediatric Cardiology, Emma Children's Hospital, Amsterdam UMC Location University of Amsterdam, Amsterdam, The Netherlands, 4 Department of Pediatric Immunology and Infectious Disease, Wilhelmina Children's Hospital, University Medical Center Utrecht, Utrecht, The Netherlands, 5 Department of Pediatrics, Hilversum, The Netherlands, 6 Department of Pediatrics, Willem-Alexander Children's Hospital, Leiden University Medical Center, Leiden, The Netherlands, 7 Department of Pediatrics, Medical Spectrum Twente, Enschede, The Netherlands, 8 Department of Internal medicine, Gelre Hospital, Apeldoorn, The Netherlands, 9 Department of Pediatrics, Spaarne Hospital, Haarlem, The Netherlands, 10 Department of Infectious Diseases, Leiden University Medical Center, Leiden, The Netherlands, 11 Department of Pediatric Surgery and Intensive Care, Erasmus Medical Center, Sophia Children's Hospital, Rotterdam, The Netherlands, 12 Pediatric Intensive Care Unit, Willem-Alexander Children's Hospital, Leiden University Medical Center, Leiden, The Netherlands, 13 Department of Pediatric Pulmonology, Emma Children's Hospital, Amsterdam UMC Location University of Amsterdam, Amsterdam, The Netherlands, 14 Department of Virology, Sanquin Research Institute, University of Amsterdam, Amsterdam, The Netherlands, 15 Department of Blood Cell Research, Sanquin Research Institute, University of Amsterdam, Amsterdam, The Netherlands

☯ These authors contributed equally to this work.
‡ EPB and TWK also contributed equally to this work.
* g.biesbroek@amsterdamumc.nl

**Data Availability Statement:** All relevant data are within the paper and its Supporting Information files.

## Abstract

Multisystem Inflammatory Syndrome in Children (MIS-C) is a severe inflammatory disease in children related to SARS-CoV-2 with multisystem involvement including marked cardiac dysfunction and clinical symptoms that can resemble Kawasaki Disease (KD). We hypothesized that MIS-C and KD might have commonalities as well as unique inflammatory responses and studied these responses in both diseases. In total, fourteen children with MIS-C (n=8) and KD (n=6) were included in the period of March-June 2020. Clinical and routine blood parameters, cardiac follow-up, SARS-CoV-2-specific antibodies and CD4+ T-cell responses, and cytokine-profiles were determined in both groups. In contrast to KD patients, all MIS-C patients had positive Spike protein-specific CD3+CD4+ T-cell responses. MIS-C and KD patients displayed marked hyper-inflammation with high expression of serum

**Funding:** The "Clinical features of COVID-19 in Pediatric Patients" (COPP)-study is funded by the #wakeuptocorona crowdfund initiative of the Bontius Stichting and the Leiden University Fund. Receiver of the grant EB The Kawasaki study is funded by the Dutch Foundation Kind & Handicap (The Hague, The Netherlands) and an anonymous donor through the AMC foundation. Reciever of the grant is TK. The sponsors had no role in the study design, the data collection and analysis, the writing of the report, or the decision to submit the manuscript for publication.

**Competing interests:** The authors have declared that no competing interests exist.

cytokines, including the drug-targetable interleukin (IL)-6 and IFN-γ associated chemokines CXCL9, 10 and 11, which decreased at follow-up. No statistical differences were observed between groups. Clinical outcomes were all favourable without cardiac sequelae at 6 months follow-up. In conclusion, MIS-C and KD-patients both displayed cytokine-associated hyper-inflammation with several high levels of drug-targetable cytokines.

## Background

Although coronavirus disease 2019 (COVID-19) in children is generally mild or asymptomatic [1–3], children can get seriously ill with systemic inflammation [4–9]. These children present with fever and multi-organ inflammation and dysfunction with clinical features reminiscent of Kawasaki Disease (KD) and evidence of past or recent SARS-CoV-2 exposure. This syndrome is defined as multisystem inflammatory syndrome in children (MIS-C) by the World Health Organization (WHO) [10] and the Centers for Disease Control and Prevention (CDC) [11]. Cases of MIS-C are typically observed 3-6 weeks after a local peak of SARS-CoV-2 infections in the population and there is usually a history of SARS-CoV-2 infection and/or exposure prior to symptom onset [7, 12]. About 80% of the MIS-C cases test positive for anti-SARS-CoV-2 antibodies, and a minority test positive for SARS-CoV-2 by RT-PCR [12, 13], supporting the notion that SARS-CoV-2 triggers this disease.

KD and Kawasaki Disease Shock Syndrome (KDSS), a more severe presentation of KD, are well-known inflammatory diseases in children. The etiology of KD is not yet fully understood, although a viral trigger is likely, given its seasonal occurrence and reported associations with several viruses, including non-pandemic coronaviruses [14–16]. Despite the similarities, there are notable differences in clinical presentation between MIS-C and KD patients. Children with MIS-C are generally older than children with classic KD and present more often with shock and marked myocardial dysfunction. Furthermore, gastrointestinal symptoms are more frequently observed and MIS-C patients sometimes lack typical signs of KD (i.e. mucosal involvement and/or strawberry tongue, conjunctivitis, rash, erythema and edema of hands and feet and cervical lymphadenopathy) [4–9]. Given the severity of both diseases, it is important to evaluate the immune cascade in both diseases to potentially optimize treatment and outcome. Hence, we compared clinical and inflammatory responses in children with MIS-C and children with KD.

## Methods

### Patients

Two prospective cohort groups included between March-June 2020 were studied: one group consisted of eight MIS-C patients, the other of six KD patients.

Patients with MIS-C were included in a nationwide cohort of SARS-CoV-2 infected pediatric patients (Clinical features of COVID-19 in Pediatric Patients" (COPP)-study, www.covidkids.nl). All these patients met the CDC [11] or WHO criteria for MIS-C [10] (S1 File). Patients with KD were enrolled in the ongoing study 'Etiology, course and long-term effects of Kawasaki Disease' (NL41023.018.12) when they had fever and fulfilled additional clinical criteria compatible with KD as defined by the American Heart Association [17].

Clinical and cardiac follow up of patients was performed at the pediatric cardiac outpatient clinic at the Amsterdam Medical University Center, Willem-Alexander Children's Hospital, Leiden University Medical Center or Sophia Children's Hospital, Erasmus MC by one of the

pediatric cardiologists and pediatric immunologists within approximately eight weeks after admission. Cardiac imaging was performed by echocardiography and *z* values were derived from the provided data.

Clinical and laboratory information was provided by their regular clinician by online anonymous information sheets and processed in a combined database (Castor).

All patients were included in accordance with study protocol and with the European Statements for Good Clinical Practice, which includes the provisions of the Declaration of Helsinki of 1989. Informed consent and written approval were obtained from all parents and children older than 12 years of age. The Medical Research Ethics Committee (MREC) of the Amsterdam UMC provided ethical approval for the Kawasaki study with the reference number: 2012_155. For the COPP-study no ethical approval was deemed necessary by the Medical Research Ethics Committee of Leiden UMC.

## Serum protein analysis

Cytokine levels and additional biomarkers in serum were measured with the Bio-Plex system and analyzed with the BioPlex Manager software version 6.2 (Bio-rad Laboratories, Hercules CA, USA). A total of 105 unique serum proteins were measured (S1 File) of which 89 serum proteins were above the detectable range and available for analysis. Some serum proteins were measured more than once. If this was the case, measurement with the best technical performance (the least 'out of range' measurements) was used for analysis. Samples taken in the acute phase of the illness were tested in six MIS-C patients and four KD patients. All samples were taken prior to treatment, except for one who had already received intravenous immunoglobulin treatment. Follow-up samples (ranging from 2-9 weeks after presentation) were available from five MIS-C patients and five KD patients. Samples of thirteen healthy adults were available as pre-pandemic laboratory controls.

## In vitro T-cell stimulation assay and flow cytometric analyses

For more detailed methods we refer to previous reports [18] and S1 File. In short, peripheral blood mononuclear cells (PBMCs) of six MIS-C patients and four KD patients were incubated with SARS-CoV-2 peptide pools at 37°C (5% $CO_2$). PBMCs were derived from samples taken at median 10 days (range 5-15 days) after disease onset in the MIS-C patients, and at median 42 days (range 8-63 days) for the KD patients. Activation with soluble αCD3 (Thermofisher, clone HIT3A) was used as a positive control and unstimulated cells (no addition of peptides) where used as negative control. Flow cytometric analyses of the PBMCs were performed with combinations of antibodies as described in the S1 File.

## Statistical analyses

Statistical analyses were performed in IBM SPSS Statistics 25. Non-parametric methods were used to compare groups because of a small sample size and a non-normal distribution (Mann-Whitney-U and Kruskal-Wallis tests). Correction for multiple testing was performed using the two-stage step-up method of Benjamini, Krieger and Yekutieli with a Q-value set at 1%. Agglomerative hierarchical clustering was performed using average linking (Between Groups) with squared Euclidean distance. Graphs were made in GraphPad Prism 8.4.2.

## Results

A total of eight MIS-C patients and six KD patients were included in the period of March-June 2020.

## Clinical and laboratory characteristics

The patients were admitted to ten different hospitals in the Netherlands. The median length of stay was nine days (range 5-12 days) for MIS-C patients and six days (range 5-19 days) for KD patients.

Patient characteristics are depicted in Table 1 (per patient in S1 Table). Remarkable, but not significantly different were the age (13 year, range 7-19 years in MIS-C patients, versus 5 years, range 2-16 years, in KD patients), incidence of shock (n=5/8, 63% versus n=2/6, 33%, respectively), and PICU admissions (n=7/8, 88% versus n=2/6, 33%, respectively). In total five MIS-C children displayed features of complete (3/8) or incomplete (2/8) KD (S1 Table).

In two MIS-C cases, prominent coronary arteries (S1 Table, Z-score +2,7 and +4) and in two cases decreased myocardial function (EF < 50 or FS < 28) was observed, compared to one case of myocardial dysfunction in the KD group. Five out of the eight MIS-C cases had shock, which was classified as cardiogenic shock in two and distributive shock in three patients. Macrophage activation syndrome (MAS) complicated MIS-C in four out of eight patients, whereas MAS was observed in none of the KD patients.

Two MIS-C children tested positive for SARS-CoV-2 by RT-PCR on nasopharyngeal aspirate or swabs. All MIS-C cases tested positive for total SARS-CoV-2 antibodies. All KD cases tested negative for total SARS-CoV-2 antibodies as well as RT-PCR.

In 5 out of 8 MIS-C patients, consecutive SARS-CoV-2 antibody samples were taken from admission up to 155 days after disease onset (S1 Fig). Overall antibody titers dropped in the seropositive patients (5/8) over time, total antibody levels were still detectable for all five patients and IgM only for one.

**Clinical course after admission.** A total of six out of eight MIS-C children were given intravenous immunoglobulin (2 g/kg) in the first 24-48 hrs after hospital admission (Table 1). In addition, five patients received high dose steroids. All received initial broad-spectrum therapeutic antibiotic coverage until cultures were negative. Subsequently, three children were given high dose aspirin and four children low-weight molecular heparin. In seven patients cardiac and clinical follow-up took place up to 155 days after discharge. In all cases the myocardial function normalized, and no residual cardiac damage or coronary abnormalities were observed using echocardiography (Table 1). Follow-up was further initiated as per protocol in KD in our centers with visits 6 and 12 months after discharge.

*Cellular counts and responses to SARS-CoV-2.* Total lymphocyte and neutrophil counts, CD3$^+$CD4$^+$ and CD8$^+$ T-cell proportions, including frequencies of effector (CD45RA$^+$CD27-) and naïve (CD45RA+CD27+) T-cell phenotypes and granzyme B (GMZB) detection in CD3$^+$CD8$^+$ (p=0.07) did not significantly differ between groups (Fig 1 and S2 Fig). GMZB detection in CD3+CD4+ T-cells was significantly higher in MIS-C compared to KD patients (p=0.02, S2 Fig).

To identify whether the KD patients were responding to SARS-CoV-2, PBMCs of KD and MIS-C patients were stimulated with a SARS-CoV-2 S-peptide pool (S1 File). The S-protein was recognized by CD3+CD4+ T cells in all children within the MIS-C group and responded, which contrasted with the total lack of such responses in the seronegative KD group (n=4). The median percentage of SARS-CoV-2 S-peptide-pool-reactive CD4 T-cells was 0.031% (range 0.016-0.038%, exemplified in Fig 1, panel D). Stimulated cells displayed IFN-γ (Fig 1, panel D and E).

**Cytokine profiles.** 59 of 81 tested serum proteins were significantly increased in the acute phase in children with MIS-C or KD as compared to healthy controls (Fig 2, S3 Fig, S2 Table). Most (53 of 81) serum protein levels were higher on average in the acute phase of illness in

children with MIS-C (n=8) than in children with KD (n=4), but this was not statistically significant after correction for multiple testing. At follow-up (between 2 and 9 weeks after presentation), serum levels of inflammatory proteins had decreased but had not completely normalized, both in patients with MIS-C (n=6) and KD (n=5).

## Discussion

In this case series from the Netherlands, we describe the clinical characteristics and inflammatory markers of children that presented with MIS-C and KD in the first wave of the SARS-CoV-2 pandemic. We observed different clinical presentations within the MIS-C group consistent with the clinical heterogeneity of MIS-C patients described previously [19]. In a CDC report of 570 patients in the USA several subgroups of MIS-C patients were identified [19]. One group of patients had high degree of organ involvement and high prevalence of shock, and another group of patients more commonly met the criteria for complete KD and had higher prevalence of coronary artery aneurysms and dilatation and lowest prevalence of complications such as shock and temporary cardiac dysfunction [19]. SARS-CoV-2 seems to trigger both of these phenotypes. In our cohort, we observed a decrease in SARS-CoV-2 IgM antibodies over time and a Spike protein-specific CD3+CD4+ T cell response in both groups of MIS-C patients.

Given the overlapping feature of MIS-C with KD and KDSS, an important question remains to what extent the pathophysiology of KD/KDSS and MIS-C share commonalities. In both MIS-C and KD patients, we observed a very high presence of many pro-inflammatory serum cytokines such as IL-6, sTNFR1&2 and the IFN-γ induced chemokines CXCL 9, 10 and 11. Even so, stimulation of CD3+CD4+ T cells of MIS-C patients with SARS-CoV-2 S-peptide pools led to activation and production of IFN-γ. Also in other cohorts of MIS-C patients increased cytokine levels were observed [20, 21], and more specific dysregulation of the IFN-γ pathway [22, 23]. IL-6 and CXCL10 levels were more increased in KD patients compared to MIS-C patients in a paper of Consiglio et al [24]. Likewise, IL-6 and IFN-γ are putative biomarkers in children with KDSS [25] and children with both MIS-C and KDSS have been successfully treated with IL-1 (anakinra) and IL-6 (tocilizumab) receptor antagonists [12, 14, 26–28]. The remarkable overlap in clinical and laboratory features between KDSS and MIS-C is intriguing [29–37]. Larger cohort studies are needed to understand which inflammatory cascades are involved in both diseases.

All MIS-C children in our cohort recovered with a good response to treatment with IVIG alone or in combination with steroids, although several MIS-C patients have been reported to require biologicals [12, 38]. We did not observe cardiac sequelae in the MIS-C patients at the follow-up visits 2-3 months after the disease episode. Nevertheless, further imaging with late enhancement MRI is needed to accurately detect the presence of residual cardiac damage [39].

Only several small studies have compared immune signatures in both diseases, and this study adds to this comparison by its prospective and coinciding recruitment of both patient groups that allowed for similar clinical and laboratory data assembly. Another strength of this study is the longitudinal follow-up of children into the convalescence phase. There are also several limitations to our study, most importantly the limited sample size that might have influenced our ability to detect significant differences between groups. Similarly, patients within the groups were clinically heterogeneous and in combination with the small sample sizes, subgroup analyses could not be performed. There is a need for larger cohort studies and more in-depth immune analyses to better understand the etiology of SARS-CoV-2 associated inflammatory syndromes compared to other auto-inflammatory syndromes.

In conclusion, our study shows marked hyper-inflammation with high levels of drug-targetable cytokines in both MIS-C and KD patients. Although we did not observe significant

differences between groups, these finding can potentially guide optimal treatment interventions in the future in children with both these diseases.

## Supporting information

**S1 File. Supplemental methods.**
(PDF)

**S1 Table. Patients characteristics.**
(PDF)

**S2 Table. Serum proteins levels, 2Log fold expression compared to healthy controls.**
(PDF)

**S1 Data. Immune cell phenotyping data.**
(XLSX)

**S2 Data. Serum proteins levels data.**
(XLSX)

**S3 Data. SARS-CoV-2 titers data.**
(XLSX)

**S1 Fig. SARS-CoV-2 antibody responses over time in children with MIS-C.** (A) Total (IgA, IgM, IgG) and IgM Spike-protein specific antibody titers measured in serum and calculated in OD/CO ratios over time in 5 MIS-C patients. (B) Total antibody titers in the MIS-C patients in relation to the time serum was obtained after admission. Antibodies were considered positive if the OD/CO ratio was > 1. Abbreviations: OD = optical density; the read out value of the antibody assay signal, CO = cut off; the threshold value for an antibody signal to be 'positive'.
(TIF)

**S2 Fig. CD3+ T-cell subsets per patients and groups.** (A) C3+CD4+ subset T-cell counts depicted per patient, MIS-C, ICU stay and age. (B) C3+CD8+ subset T-cell counts depicted per patient, MIS-C, ICU stay and age. (C) CD3+CD8+ effector T-cell percentages in MIS-C (+) and KD patients (-). (D) CD3+CD8+ effector T-cell percentages depicted for patients that stayed at the ICU (+) or not (-). (E) CD3+CD4+ effector T-cell percentages in MIS-C (+) and KD patients (-). (F) CD3+CD4+ effector T-cell percentages depicted for patients that stayed at the ICU (+) or not (-). (G) percentage of CD3+CD8+ GZMB positive T-cells (cytotoxic T-cells) in MIS-C (+) and KD patients (-). (H) Percentage of CD3+CD4+ GZMB positive T-cells in MIS-C (+) and KDe patients (-). (I) Total lymphocytes counts in MIS-C (+) and KD patients (-). (J) Total neutrophil counts in MIS-C (+) and KD patients (-). Abbreviations: Te = effector T-cells, Tem = effector memory T-cells, Tcm = central memory T-cells, Tn = naïve T-cells, GZMB = Granzyme B.
(TIF)

**S3 Fig. Violin plots of serum cytokines across study groups.** Violin plots of the highest log fold increased serum cytokines compared to healthy controls, and all represented in the lowest hierarchical cluster in Fig 2.
(TIF)

## Acknowledgments

We thank all children and parents that participated in this study. Furthermore, we gratefully acknowledge all study and laboratory staff and collaborating institutes for their dedication to this project. We acknowledge the help of S. van Veen and P. Ruibal at the Department of Infectious Diseases, Leiden University Medical Center, in determining cytokine profiles. KIRI group consists of D. Schonenberg-Meinema, A. M. Tutu van Furth, M. Boele van Hensbroek and H. Scherpbier, supporting direct patient care and logistics.

## Author Contributions

**Conceptualization:** G. Biesbroek, B. Kapitein, I. M. Kuipers, M. P. Gruppen, M. van der Kuip, J. M. van den Berg, E. P. Buddingh, T. W. Kuijpers.

**Data curation:** B. Kapitein, I. M. Kuipers, M. P. Gruppen, D. van Stijn, T. E. Peros, M. van Veenendaal, M. H. A. Jansen, C. W. van der Zee, M. van der Kuip, E. G. J. von Asmuth, M. G. Mooij, M. E. J. den Boer, G. W. Landman, M. A. van Houten, D. Schonenberg-Meinema, A. M. Tutu van Furth, M. Boele van Hensbroek, H. Scherpbier, N. Ketharanathan, M. Blink, C. L. H. Brackel, H. L. Zaaijer, P. Hombrink, J. M. van den Berg, E. P. Buddingh, T. W. Kuijpers.

**Formal analysis:** G. Biesbroek, D. van Stijn, T. E. Peros, H. L. Zaaijer.

**Funding acquisition:** E. P. Buddingh, T. W. Kuijpers.

**Investigation:** I. M. Kuipers, P. Hombrink.

**Methodology:** G. Biesbroek, B. Kapitein, I. M. Kuipers, D. van Stijn, K. E. van Meijgaarden, T. H. M. Ottenhoff, S. A. Joosten, E. P. Buddingh.

**Project administration:** T. E. Peros, T. W. Kuijpers.

**Supervision:** B. Kapitein, I. M. Kuipers, E. P. Buddingh, T. W. Kuijpers.

**Visualization:** G. Biesbroek.

**Writing – original draft:** G. Biesbroek, B. Kapitein, E. P. Buddingh, T. W. Kuijpers.

**Writing – review & editing:** G. Biesbroek, B. Kapitein, I. M. Kuipers, M. P. Gruppen, D. van Stijn, T. E. Peros, M. van Veenendaal, M. H. A. Jansen, C. W. van der Zee, M. van der Kuip, E. G. J. von Asmuth, M. G. Mooij, M. E. J. den Boer, G. W. Landman, M. A. van Houten, D. Schonenberg-Meinema, A. M. Tutu van Furth, M. Boele van Hensbroek, H. Scherpbier, K. E. van Meijgaarden, T. H. M. Ottenhoff, S. A. Joosten, N. Ketharanathan, M. Blink, C. L. H. Brackel, H. L. Zaaijer, P. Hombrink, J. M. van den Berg, E. P. Buddingh, T. W. Kuijpers.

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
