## [Editor Report · Decision Letter 0]

1 Dec 2021

PONE-D-21-31674Inflammatory responses in SARS-CoV-2 associated Multisystem Inflammatory Syndrome and Kawasaki Disease in children: an observational studyPLOS ONE

Dear Dr. Biesbroek,

Thank you for submitting your manuscript to PLOS ONE. After careful consideration, we feel that it has merit but does not fully meet PLOS ONE’s publication criteria as it currently stands. Therefore, we invite you to submit a revised version of the manuscript that addresses the points raised during the review process.

We look forward to receiving your revised manuscript.

Kind regards,

Tariq Jamal Siddiqi

Academic Editor

PLOS ONE

Journal Requirements:

3. Please amend the manuscript submission data (via Edit Submission) to include authors  D. Schonenberg, M. Boele van Hensbroek, H.J. Scherpbier, A.M. Tutu-van Furth.

5. We note that Figure Additional File 3 in your submission contain copyrighted images. All PLOS content is published under the Creative Commons Attribution License (CC BY 4.0), which means that the manuscript, images, and Supporting Information files will be freely available online, and any third party is permitted to access, download, copy, distribute, and use these materials in any way, even commercially, with proper attribution. For more information, see our copyright guidelines: http://journals.plos.org/plosone/s/licenses-and-copyright.

a. You may seek permission from the original copyright holder of Figure Additional File 3 to publish the content specifically under the CC BY 4.0 license. 

Additional Editor Comments:

Biesbroek et al. conducted an observational study on,” Inflammatory responses in SARS-CoV-2 associated Multisystem Inflammatory Syndrome and Kawasaki Disease in children”, in which they report that Multisystem Inflammatory Syndrome and Kawasaki Disease patients both displayed cytokine-associated hyper-inflammation with several high levels of drug-targetable cytokines. In my opinion this study can be improved by incorporating the following points:

1. The discussion can be improved further by discussing more about

2. When mentioning about “granzyme B (GMZB) detection did not significantly differ between groups”, the author should also state a p-value.

3. The strengths of the study need to be addressed along with how this study adds to the medical literature.

4. It can be mentioned how study focuses on a population that was not highlighted by previous authors, or whether this study demonstrates new results and how your study is superior to previous and other such studies.
---

## [Author Response · Author response to Decision Letter 0]

9 Feb 2022

10-1-2021

Dear editor,

We thank you for your considerations and the opportunity to revise our manuscript entitled ” Inflammatory responses in SARS-CoV-2 associated Multisystem Inflammatory Syndrome and Kawasaki Disease in Children; an observational study” by G. Biesbroek et al. 

Herewith we would like to submit our revised manuscript and answer to the editorial comments. You’ll find our reply underneath, discussed per point raised by the academic editor. We thank the editor for the valuable comments. The suggested edits have certainly contributed to the quality of our manuscript. 

Specific editorial comments

We have adjusted the manuscript accordingly.

The funders are enlisted in the funding information section. The Kawasaki study is funded by the Dutch Foundation Kind & Handicap (The Hague, The Netherlands) and an anonymous donor through the AMC foundation. Similarly, The “Clinical features of COVID-19 in Pediatric Patients” (COPP)-study is funded by the #wakeuptocorona crowdfund initiative of the Bontius Stichting and the Leiden University Fund. These are all non-profit organisations or sponsors. 

3. Please amend the manuscript submission data (via Edit Submission) to include authors D. Schonenberg, M. Boele van Hensbroek, H.J. Scherpbier, A.M. Tutu-van Furth.

We have adjusted the author section.

We have adjusted the methods section accordingly.

5. We note that Figure Additional File 3 in your submission contain copyrighted images. All PLOS content is published under the Creative Commons Attribution License (CC BY 4.0), which means that the manuscript, images, and Supporting Information files will be freely available online, and any third party is permitted to access, download, copy, distribute, and use these materials in any way, even commercially, with proper attribution.

We require you to either (1) present written permission from the copyright holder to publish these figures specifically under the CC BY 4.0 license, or (2) remove the figures from your submission.

We have obtained permission of the patients that the pictures can be published and can be freely accessed online. After a reminder last week we are waiting for the written permissions to be returned to us. If possible, we would like to send them as soon as possible to you upon arrival. If this is not possible, we are in agreement to remove the figure from the online materials. One patient was not in agreement and we removed the X-ray of her lungs already from the figure. 

Additional Editor Comments:

2. When mentioning about “granzyme B (GMZB) detection did not significantly differ between groups”, the author should also state a p-value.

We added the p-value for GMZB detection in the manuscript. We didn’t test significance in the CD4+ T-cell population separately, but this appeared unexpectedly to be significantly different between groups and have been added to the manuscript. 

3. The strengths of the study need to be addressed along with how this study adds to the medical literature.

The strengths of this study are the coinciding prospective recruitment of both MIS-C and KD patients and the longitudinal follow-up of children into the convalescence phase. These strengths have been added. To date, there are only a limited number of studies that compare immune signatures between MIS-C and KD and this study therefore also adds to our collective understanding of the similarities and distinct features of these two diseases.

4. It can be mentioned how study focuses on a population that was not highlighted by previous authors, or whether this study demonstrates new results and how your study is superior to previous and other such studies

The strength of this study is the prospective recruitment of patients in both disease groups which allowed for direct comparison between groups with identical study procedures and clinical comparison. Most of the performed studies have used historical Kawasaki Disease patient cohorts that might have introduced differences in clinical data assembly and stored materials. This aspect has been highlighted in the manuscript accordingly. 

We thank the editor for the opportunity to revise our manuscript and have sufficiently answered all editorial comments. 

Sincerely yours, 

Giske Biesbroek, on behalf of the authors

---

## [Decision Letter · Decision Letter 1]

21 Mar 2022

Inflammatory responses in SARS-CoV-2 associated Multisystem Inflammatory Syndrome and Kawasaki Disease in children: an observational study

PONE-D-21-31674R1

Dear Dr. Biesbroek,

We’re pleased to inform you that your manuscript has been judged scientifically suitable for publication and will be formally accepted for publication once it meets all outstanding technical requirements.

Kind regards,

Tariq Jamal Siddiqi

Academic Editor

PLOS ONE

Additional Editor Comments (optional):

Reviewers' comments:

Reviewer's Responses to Questions

**Comments to the Author**

1. If the authors have adequately addressed your comments raised in a previous round of review and you feel that this manuscript is now acceptable for publication, you may indicate that here to bypass the “Comments to the Author” section, enter your conflict of interest statement in the “Confidential to Editor” section, and submit your "Accept" recommendation.

Reviewer #1: All comments have been addressed

2. Is the manuscript technically sound, and do the data support the conclusions?

Reviewer #1: Yes

3. Has the statistical analysis been performed appropriately and rigorously? 

Reviewer #1: Yes

4. Have the authors made all data underlying the findings in their manuscript fully available?

Reviewer #1: Yes

5. Is the manuscript presented in an intelligible fashion and written in standard English?

Reviewer #1: Yes

6. Review Comments to the Author

Reviewer #1: (No Response)

7. PLOS authors have the option to publish the peer review history of their article (what does this mean?). If published, this will include your full peer review and any attached files.

Reviewer #1: No

---

## [Editor Report · Acceptance letter]

12 Apr 2022

PONE-D-21-31674R1 

Inflammatory responses in SARS-CoV-2 associated Multisystem Inflammatory Syndrome and Kawasaki Disease in children: an observational study 

Dear Dr. Biesbroek:

I'm pleased to inform you that your manuscript has been deemed suitable for publication in PLOS ONE. Congratulations! Your manuscript is now with our production department. 

Kind regards, 

on behalf of

Dr. Tariq Jamal Siddiqi 

Academic Editor

PLOS ONE